# Predicting Stable Configurations for Semantic Placement of Novel Objects

**Chris Paxton**[1]**, Chris Xie**[2]**, Tucker Hermans**[1,3]**, Dieter Fox**[1,2]
[1]NVIDIA, [2]University of Washington, [3]University of Utah

**Abstract:** Human environments contain numerous objects configured in a variety of arrangements. Our goal is to enable robots to repose previously unseen objects according to learned semantic relationships in novel environments. We break this problem down into two parts: (1) finding physically valid locations for the objects and (2) determining if those poses satisfy learned, high-level semantic relationships. We build our models and training from the ground up to be tightly integrated with our proposed planning algorithm for semantic placement of unknown objects. We train our models purely in simulation, with no fine-tuning needed for use in the real world. Our approach enables motion planning for semantic rearrangement of unknown objects in scenes with varying geometry from only RGB-D sensing. Our experiments through a set of simulated ablations demonstrate that using a relational classifier alone is not sufficient for reliable planning. We further demonstrate the ability of our planner to generate and execute diverse manipulation plans through a set of real-world experiments with a variety of objects.

**Keywords:** Deep learning for manipulation, learning for motion planning, semantic manipulation

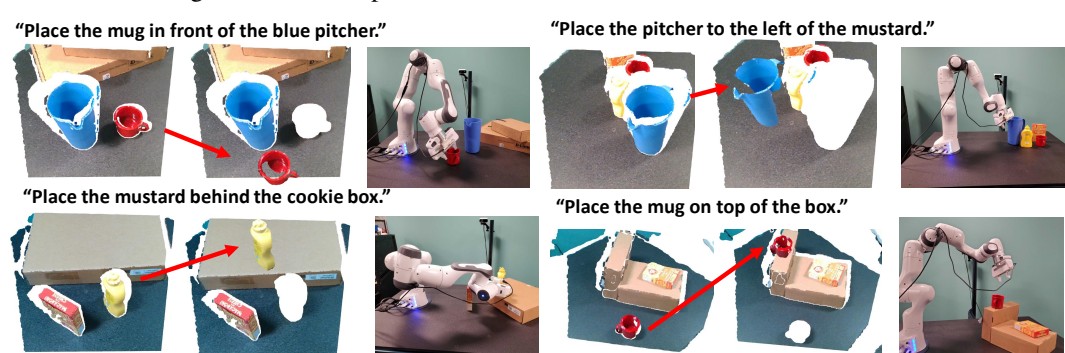

**Figure 1:** Planning to achieve different predicate relationships in the real world, given only segmented point clouds of the scene. The planner computes final placement positions and associated kinematic poses, which lets the robot generate a motion plan to place these objects at different locations in the real world.

## 1 Introduction & Motivation

Many robotics tasks in human environments involve reasoning about the relationships between different objects and their locations in a particular environment. Imagine a robot tasked with pouring a cup of coffee, it must reason about the relative position of the cup to the coffee pot, such that it pours the coffee into the cup and not onto the counter or floor. For a robot attempting to brew the coffee prior to pouring, many more multi-object relations must be reasoned about for the tasks of retrieving the necessary equipment from storage, scooping and pouring beans or grounds, and filling any required water vessels.

Classically, researchers investigating these sorts of multi-object planning and manipulation problems assume full knowledge of the objects and their poses in the scene [1]. However, for robots acting in homes and other open-world environments, it is unreasonable to assume that a deployed robot will have knowledge and accurate pose estimation of all objects in the environment. Furthermore, simple hand-engineered classifiers for determining logical predicates often exhibit unintended

5th Conference on Robot Learning (CoRL 2021), London, UK.

behavior when deployed on real-world systems [2]. For these reasons, rearrangement has recently been identified as a major challenge for robotics [3].

Given these challenges, its natural to examine the use of learning to improve task and motion planning with real sensing [4, 5, 6, 7, 8, 9]. However, previous methods fail to solve the full problem of unknown object rearrangement with physical robots. Some only operate on known objects [6, 9], others ignore or significantly restrict the space of robot control [4, 5] or relations [7, 8], while still others make assume an explicit goal configuration is given [5]. An alternative approach to solve complex manipulation tasks relies on learning model-free neural net policies instead of explicit models of conditions and effects [10, 11]. Such methods have so far failed to show the level of generalization across objects and environments capable by modern task and motion planners.

In this paper, we focus on a critical subtask of rearrangement planning–*semantic placement*–where the robot must perform a pick and place operation to move an object into a stable configuration that satisfies a desired set of semantic relations. This allows us to focus on three specific objectives not addressed in previous work on learning for task and motion planning. First, we examine the problem of planning and controlling manipulation behaviors to change inter-object relations of potentially previously unseen objects using RGB-D sensing. Second, we aim to learn the necessary relations from data to avoid the bias introduced from hand-engineered classifiers. Finally, we must learn what realistic scenes look like–so that the robot can ensure that it places objects in reasonable locations which are stable and free of undesired collisions.

Our work relies on the ability to infer inter-object relational predicates between objects proposed from an RGB-D instance segmentation, e.g. [12]. Semantic relations serve an important role in instructing robots [13]. As such, researchers have examined visual prediction of spatial relations [14, 15, 16, 17] including inference of support relations [18, 19]. While these methods can be sophisticated incorporating language [20, 21, 16] or scene graph information [22, 23, 17]; none have demonstrated the ability to integrate prediction with robot planning and control to satisfy new relations between unknown objects. Indeed, we show semantic prediction alone is insufficient for reliably planning successful semantic rearrangement.

In addition to this semantic prediction, we need to identify *physically realistic* poses where these objects can be placed. Stable placement prediction has also received attention from robotics researchers [24, 25, 26, 27]; however, learning-based approaches [24, 27] train directly to maximize stability of object placement. In contrast our approach simply distinguishes realistic configurations from invalid ones, which allows us to learn a general-purpose *scene-realism discriminator* which can capture wide distributions over realistic poses in 3D space. This provides the benefit of a more general model for use in arrangement planning, while giving up somewhat the level of precision seen in some placement-specific methods [24, 26]. In addition, it means we can use simulation-based data to train both our relational prediction and scene discriminator directly on raw point cloud and associated segmentation masks, and transfer to the real world.

We embedded our learned relational predictor and scene discriminator within a sampling-based planning framework to change relations in the scene in a goal directed way. By planning directly over changes in segment pose in a point cloud, we can decouple the goal generation problem from the robot control problem, allowing us to leverage model-based, state-of-the-art motion planning and grasp prediction algorithms [28, 29] to perform the necessary manipulation. We further accelerate our planner by learning an object pose sampler conditioned on desired relations to initialize the optimization, similar to the grasp planning approach in [28]. Production of these goal states can then be used for semantically-defined placement tasks.

We highlight the advantages of our approach over a variety of baselines and ablations of our full method. Critically, we demonstrate that both the relational classifier and scene discriminator are necessary for reliably generating successful plans. We then demonstrate our approach in the real world using a Franka robot (Fig. 1). Our experiments constitute the first physical-robot demonstration that combine learned models for inter-object relations and stability estimation enabling rearrangement of novel objects. Crucially, we show that the combination of relationship classifier and scene discriminator allows us to plan placements for a variety of relationships in cluttered scenes. In addition, our models are trained entirely in simulation with no need for real-world fine tuning.

## 2 Methods

Given a single view of a scene containing objects for which our robot potentially has no previous experience, we wish for the robot to rearrange the scene to satisfy some new set of logical constraints. For example the robot may be tasked to move the *query object* $i$ to be on the far right side of *anchor object* $j$ or to be stacked on top of object $k$. Each individual relationship between $i$ and $j$ is referred to as a predicate $\rho_{ij}$; we can describe multiple logical relationships as the vector $\vec{\rho}_{ij}$.

We assume we are given a partial-view point cloud $Z$ with segment labels for each point to identify the different objects. Given this point cloud and a set of logical predicates describing the desired relationships $\vec{\rho}_{ij}$, the robot must find a pose offset $\delta$ (3D translation and planar rotation) for object $i$ that satisfies $\vec{\rho}_{ij}$ and is additionally a stable, physically valid placement pose in the environment.

Thus there are two key components in our rearrangement and placement planning approach: predicting which poses objects can be physically placed and predicting which poses satisfy the given high-level instructions. We formalize predicate planning as the following problem:

$$\underset{\delta}{\text{argmin}} \quad c(\delta) = \lambda_f \|1 - f(x_i \oplus \delta, Z')\|_2 + \lambda_\rho \|p_\rho(x_i \oplus \delta, x_j) - \vec{\rho}_{ij}\|_2 \tag{1}$$

$$\text{subject to} \quad T(Z, x_i \oplus \delta) = Z' \tag{2}$$

$$f(x_i \oplus \delta, Z') > \epsilon_f \tag{3}$$

$$p_\rho(x_i \oplus \delta, x_j)\,[\vec{\rho}_{ij}] > \epsilon_\rho \tag{4}$$

$$\Pi(x_i \oplus \delta) = 1 \tag{5}$$

where $x_i$ and $x_j$ are the object point clouds for objects $i$ and $j$, respectively, and $\oplus$ denotes the application of the 3D translation and planar rotation. At the heart of our planning cost, Eq. 1, are two models. The first $f(x_i \oplus \delta, Z')$ is a neural net trained to determine if the resulting scene is physically realistic and stable. The second term defines the cost associated with matching the set of target predicates, where $p_\rho(x_i \oplus \delta, x_j)$ estimates the set of predicate relationships between the transformed point cloud $x_i \oplus \delta$ and $x_j$. This cost implies maximizing the likelihood that the resulting scene is both realistic and satisfies the desired predicates.

In addition to the cost function, we put minimum bound constraints on the physical feasibility (Eq. 3) and predicate probabilities (Eq. 4). We use $\vec{\rho}_{ij}$ as an index in Eq. 4 to extract the subset of predicted relations that must be satisfied. This ensures we never attempt to plan to a scene configuration with low probability of success, even if it defines a local optimum of the objective. Eq. 2 models the transition of applying offset $\delta$ to $x_i$ in the observed scene $Z$ to generate the resulting scene $Z'$, which we evaluate in the cost and other constraints. Finally Eq. 5 ensures that sampled object offsets are visible in the camera view of the robot, since our cost and constraint evaluations would be ill-defined otherwise. We describe the details of these models and their construction in Sec. 2.1 and give a detailed description of our data generation process for training in Section 2.3.

We use a variant of the cross-entropy method (CEM) [30] to solve this constrained optimization problem. CEM has previously been applied to robot motion planning [30], including semantic motion planning from learned models [31]. We provide further details of our planner in Section 2.2.

### 2.1 Rearrangement Planner Models and Training

We train multiple neural networks to compute the values needed to instantiate our planning problem defined by Equation 1: the current set of predicates, and the discriminator score which describes whether or not a particular set of object points $x$ defines a realistic configuration in the scene.

**Object and scene encoders** The core piece of the model is a PointNet++ [32] encoder which extracts a lower-dimensional object representation $h$. Each network takes two objects $i$ and $j$ represented as point clouds $x_i$ and $x_j$, as well as the scene point cloud $Z$, as input. Given an observation point cloud $x$, we learn a mapping $e(x) \rightarrow h$ for the objects, in order to get two latent representations $h_i$ and $h_j$ for the query and anchor objects. The object encoder is a Pointnet++ model with three set abstraction layers; see the supplemental material for further details.

We train a separate scene encoder $e_Z(x_Z)$ to capture the objects' relation to other scene geometry when predicting where it can be placed. This outputs $h_Z$ encoding scene-specific information, where $x_Z$ is centered around the anchor point $o_j$ – the centroid of $x_j$. Points are sampled in a radius of

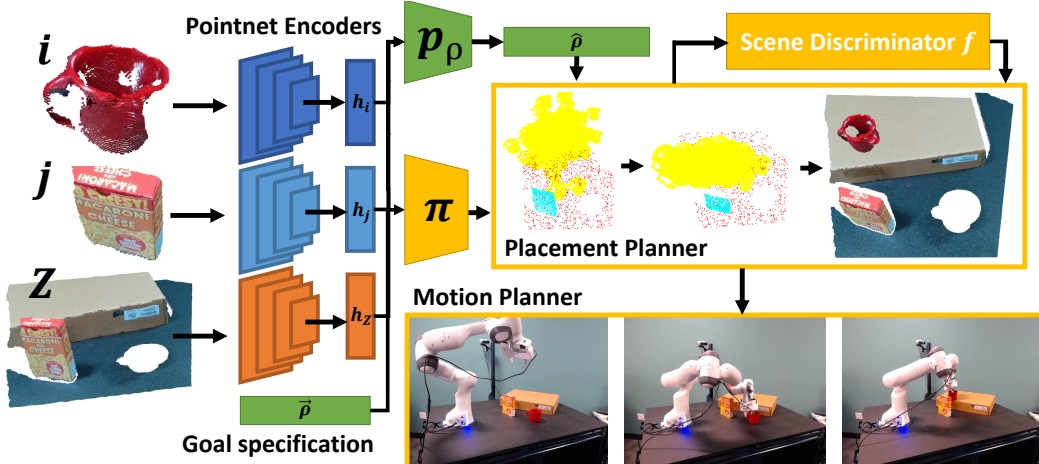

**Figure 2:** System for rearrangement of unknown objects according to spatial relations, where we wish to move a query object $i$ (in this case a red mug). We encode objects with Pointnet++ to predict predicates, based on the region around an "anchoring" object, $j$ (the macaroni box) to which the query is relative. Query locations are sampled based on a learned prior $\pi$, and are classified by the scene discriminator network $f$ as either realistic or unrealistic. This is used as a part of an optimization algorithm to find a stable, kinematically feasible query pose so that we can place the query object $i$ in a new location.

$r = 0.5$m around $o_j$. We choose this representation to define features from the fixed perspective of the anchoring object $j$.

**Relation predictor** The predicate classifier network $p_\rho(x_i, x_j) \rightarrow \hat{\rho}_{ij}$ estimates which predicates are true for a pair of object point clouds. It uses the representations from the object encoder, $e(x_i) = h_i$ and $e(x_j) = h_j$, for the query and anchor objects $i$ and $j$, respectively. These are passed into an MLP which predicts a vector of length $N_{predicates}$. When used as part of the planning algorithm, we ignore predicted predicates that are not part of the goal specification.

**Pose prior** In addition, we learn a prior distribution over possible poses where the object might satisfy the predicates, relative to the anchoring object. We implement this prior $\pi(x_i, x_j, Z, \vec{\rho}_{ij}) \rightarrow \{\alpha_k, \mu_k, \sigma_k\}_{k=1:K}$ as a Mixture Density Network (MDN) [33], which predicts the parameters of a Gaussian mixture model with $K$ components. This GMM distribution defines the probability of a specific pose offsets $\delta$ with respect to the anchor point $o_j$. The relationships are assumed to be spatially defined relative to this object, so that the predicted center of the query object $i$ is $\hat{o}_i = o_j \oplus \delta$. Once trained we can produce samples from the MDN by evaluating it for the current observations and then applying standard GMM sampling using the output parameters.

As with the relation predictor $p_\rho$, the pose prior uses the object encoder to get lower-dimensional representations $h_i$ and $h_j$ for each object as well as the scene encoder to predict $h_Z$.

**Scene discriminator** The relation predictor and prior on their own are not enough to find stable placement poses. As such, we define a discriminator network $f(x, Z)$ trained to predict if a given configuration results in a placement that is physically realistic w.r.t. the training data, i.e. it is stable.

This model uses a slightly different architecture from the above models, and unlike them we first center the scene on $\hat{o}_i$, the query point and potential new pose for object $i$. We look at the local region around $\hat{o}_i$ with a fixed radius $r = 0.5$m to classify whether a particular $\delta$ would result in a realistic placement pose. We use a single PointNet++ model to encode points from both objects together, given a label indicating which points belong to the query object (the object whose placement we are attempting to find). In practice, the sphere-query necessary to extract the local context around a particular pose is the same as that used in PointNet++, so this operation can be performed quickly at inference time.

**Transform operator** We require one additional operator to plan on point clouds: the transformation operator $T(Z, x_i \oplus \delta)$. We assume a deterministic transition function and rigidly transform points associated with the query object according to the relative pose offset $\delta$ constructing a new scene $Z'$.

**Model Training** We jointly train the object and scene encoders $e$ and $e_Z$, relationship predictor $p_\rho$, and the pose prior $\pi$. The relationships can be directly supervised from our training data, given

knowledge of the ground truth predicates, and are trained with a binary cross entropy loss. The prior $\pi$ is trained to predict the offset $\delta$ from the anchor point $o_j$ to the observed pose of the query object, $o_i$, in the training data; i.e. $\delta = o_j \ominus o_i$.

To train the scene discriminator, we first note that all of our training data consists of stable object placements, thus there is no negative data nor stability supervision available. Instead, we create negative data online by applying random $\delta$ offsets to the pose of the query object $i$. We simply sample a random $\delta$ and apply our transform operator to create a new scene $Z^- = T(Z, x_i \oplus \delta)$. These $\delta$ were sampled to be between 2 and 15 centimeters in a random direction. The resulting scenes are highly likely to not be physically realistic nor stable (e.g. object $i$ is floating or is in collision).

---

**Algorithm 1** Placement planning algorithm pseudocode.

---

1: **function** FINDPLACEMENT(object point cloud $x_i$, object point cloud $x_j$, scene $Z$, goal $\vec{\rho}_{ij}$)
2:     **for** $s \in \text{range}(1, N)$ **do**
3:         $\vec{\delta}_s = \emptyset$
4:         **while** $\text{length}(\vec{\delta}_s) < B$ **do**                                   ▷ Rejection Sampling
5:             **if** $s = 0$ **then**
6:                 $\delta \sim \pi(x_i, x_j, Z, \vec{\rho}_{ij})$         ▷ Sample initial poses from learned prior
7:             **else**
8:                 $\delta \sim \mathcal{N}(\mu', \Sigma')$     ▷ Sample subsequent poses from surrogate distribution
9:             $Z' \leftarrow T(Z, x_i \oplus \delta)$              ▷ Shift object point clouds by $\delta$
10:            `realistic` $\leftarrow f(x_i \oplus \delta, Z') > \epsilon_f$        ▷ Determine if pose is realistic
11:            `goal` $\leftarrow p_\rho(x_i \oplus \delta, x_j)[\vec{\rho}_{ij}] > \epsilon_\rho$     ▷ Classify if goal predicates are true
12:            `in_view` $\leftarrow \Pi(x_i \oplus \delta)$            ▷ Ensure it will be visible
13:            **if** `realistic` and `goal` and `in_view` **then**
14:                 $\vec{\delta}_s \leftarrow \vec{\delta} \cup \delta$               ▷ Add new $\delta$ to batch of samples
15:         Compute cost $c(\delta)$ for each $\delta \in \vec{\delta}_s$
16:         Sort $\vec{\delta}_s$ and take top $N_{\texttt{elite}}$
17:         Fit surrogate distribution parameters $(\mu', \Sigma')$
18:     **return** lowest-cost $\delta$ seen so far, or $\emptyset$

---

## 2.2 Manipulation Planning with Relationship Models

We now describe how we solve the optimization problem for Equation 1 using the components defined above. Alg. 1 describes how we can place an object so as to satisfy a particular relationship. We take as given a set of desired relationships $\vec{\rho}_{ij}$ in scene $Z$, and target objects $i$ and $j$, where $i$ is the *query object* that we will be moving and $j$ is the *anchor object* that will be kept stationary.

Initially, we perform rejection sampling to draw a batch of $B$ candidate pose offsets from our MDN prior $\pi(x_i, x_j, Z, \vec{\rho}_{ij})$ (line 6) keeping only those that satisfy the full set of constraints (lines 10–14). We sample until either a time budget has been reached or $B$ samples have been successfully drawn.

For each sample $\delta$ we compute a new query point cloud using the transform function $Z' = T(Z, x_i \oplus \delta)$ (line 9) and thus satisfy the constraint in Eq. 2 by construction. We then evaluate the remaining constraints in Eq. 3–5 (lines 10–12) accepting only feasible samples (lines 13–14). Next, as per the cross-entropy method, we evaluate the cost of the valid samples (line 15) and fit a surrogate distribution to the best scoring $N_{\texttt{elite}}$ samples with mean $\mu'$ and variance $\Sigma'$ (lines 16–17). At each subsequent step, we draw $B$ samples $\delta \sim \mathcal{N}(\mu', \Sigma')$ from our current surrogate distribution, compute the scores again, and re-sample, until we have performed $N$ sampling iterations.

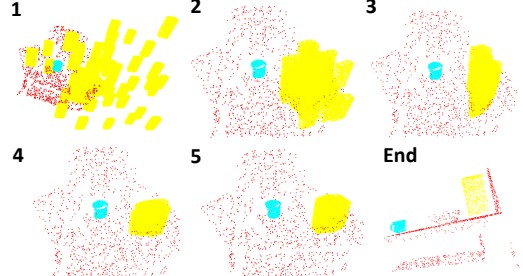

**Figure 3:** Depiction of the planning sequence. Initial samples are drawn from an MDN prior $\delta \sim \pi(Z, \vec{\rho}_{ij})$. Final poses satisfy both realism and predicate constraints finding a reasonable placement pose.

Fig. 3 gives an example for how the planning algorithm works in practice to achieve the goal that the yellow object be aligned to the right of the cyan object. In both Fig. 2 and Fig. 3, yellow objects are predicted positions of the query object; cyan represents the anchor object, and red points represent downsampled scene geometry. This is an accurate depiction of the inputs into our model. The final frame of Fig. 3 has been rotated to show the precise alignment with the table surface.

### 2.3 Dataset Creation

We generated a large-scale dataset in simulation of RGB-D images with associated segmentation masks and relational predicates. We provide binary labels between all visible pairs of objects for all predicates listed in Table 2. Each scene consists of 3 to 7 random Shapenet [34] objects in stable configurations on various surfaces, including in stacks. We include mugs, bowls, and bottles, as well as boxes and cylinders of random sizes. Examples of generated scenes are shown in the supplemental material. Objects were placed in random configurations, and we ran physics forward to find stable arrangements. We then rendered images both with and without each object. In order to train the rotation model, we render each object on its own and apply a random rotation. Directional predicates (*left of*, *right of*, etc.) are computed based on bounding box overlap; others are computed based on distance between meshes or ground-truth position and orientation. See the supplementary materials for further details.

## 3 Experiments

We first performed a set of simulation experiments on scenes that resembled our real world objects examining different versions of our model. First, we show an ablation test of our method on held-out YCB objects [35], with a set of known grasps from Eppner et al. [36], in order to show that our algorithm with discriminator is better able to find stable positions for objects and scenes that did not appear in our training data. Second, we show a break-down of the predicate results, showing that our learned model is comparable or better at capturing a wide range of difficult relationships even in partially-occluded scenes.

We sampled 100 random scenes in the kitchen environment from Fig. 4, with objects either positioned on top of the counter or in the top drawer. These objects did not appear in the training set. Table 1 shows a breakdown of several different versions of the planner after running experiments on 100 random scenes, each with a random predicate goal chosen from (in front, behind, left, right). Due to the random placements of the objects, not all scenes are feasible, and in many cases the goal pose would be occluded or off of the table to the front, resulting in challenging planning problems.

| Variant | Found Predicates | Found Realistic | Successful | Stable Pose |
|---|---|---|---|---|
| Full model, $\lambda_f = 100$ | 93 | 87 | 84 | 71 |
| Full model, $\lambda_f = 1$ | 94 | 81 | 78 | 67 |
| No Discriminator ($\lambda_f = 0$) | 100 | 3 | 3 | 25 |
| Mean only | 96 | 27 | 27 | 39 |
| MDN Prior | 99 | 6 | 6 | 42 |

**Table 1:** Comparison between different versions of the planner, when tested on 100 random multi-object scenes in a kitchen environment. "Stable pose" is when the object center moved less than 5 cm after placement.

The different baselines look at the effects of the discriminator model, which determines whether or not a scene is realistic and whether or not an object can be placed at a particular pose. For example, the "no discriminator" case does not use the discriminator at all, and is very good at finding poses matching the predicate goal but not finding stable poses. We also vary the weight of the discriminator $\lambda_f$ in several examples. For these experiments we use a batch size, $B$, of 100.

We ran two experiments that do not use the discriminator in our "full" planning approach. **Mean only** draws samples only from the mean of the MDN prior $\pi$. This looks at what performance is like with a learned policy. **MDN Prior** uses the learned mixture density function as a cost function in place of using the discriminator, since presumably this might capture much of the same information. We can see that it actually does a fairly good job at matching the predicates, but is not very discriminative when it comes to finding stable poses for placement. Both of these perform notably worse at finding stable poses in our test environments.

|          | Left of | Right of | In Front | Behind | Above | Below | Near | Touching | Centered |
|----------|---------|----------|----------|--------|-------|-------|------|----------|----------|
| Learned  | 0.92    | 0.93     | 0.74     | 0.65   | 0.90  | 0.92  | 0.88 | 0.90     | 0.70     |
| Baseline | 0.95    | 0.95     | 0.67     | 0.87   | 0.91  | 0.94  | 0.90 | 0.31     | 0.01     |
| %True    | 13.9%   | 14.0%    | 4.6%     | 5.0%   | 4.3%  | 4.3%  | 29.0% | 12.7%   | 5.6%     |
| %False   | 86.1%   | 86.0%    | 95.4%    | 95.0%  | 95.7% | 95.7% | 71.0% | 96.3%   | 84.4%    |

**Table 2:** F1-score of the predicate predictor $p_\rho$ in held-out randomly-generated simulated test scenes. Some predicates in our scenes can be very difficult due to clutter and occlusions, but our learned models are either on par with or better than most all baselines. Bottom two rows show prevalence in the evaluation data set.

### 3.1 Scene Discriminator Performance

Here, we examine the performance of our scene discriminator. To do this, we compare placement poses sampled from the MDN prior distribution in randomly-generated kitchen scenarios to placement scores after optimization. We place the object at the new pose and then run 500 simulation steps to allow the object to settle into its final pose. We then compare the discriminator's confidence score with how much the object moved. For these experiments we generated 100 random scenes, each with a random predicate goal so as not to bias it to a particular subset of the problem space. We ignore scenes if no feasible pose was found according to the discriminator.

We found that in 95% of these scenarios, the discriminator was able to find a stable pose to place a particular object, and in 90% of all scenarios the planner was also able to match the specified predicates. This shows that not only can we find realistic positions, but that the discriminator does not preclude achieving specified goals. Note the higher perceived success rates than in the planner comparison in Table 1: this is because we only test scenes where the planner was initially confident that it could find a solution.

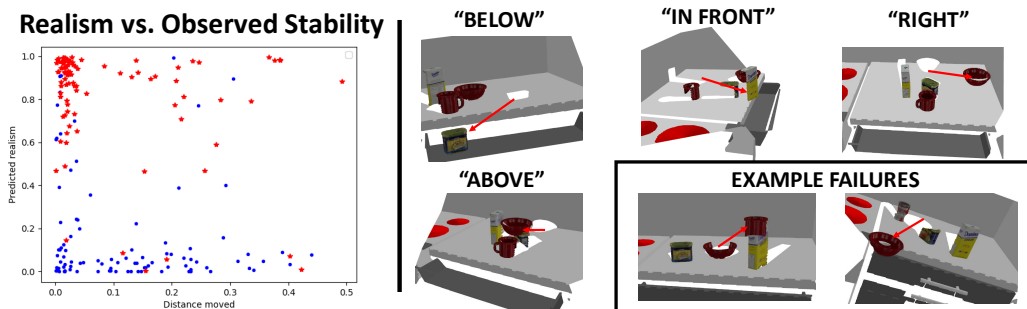

**Figure 4:** Sim experiment results. **Left:** correlation between predicted realism (y axis) and distance moved after placement (x axis). The blue dots show randomly sampled poses, while the red stars indicate poses after running the placement planner. More realistic poses are much more stable than less realistic ones, although in some cases unrealistic poses do not move much either. **Right:** selected successful and unsuccessful simulation results. Images show point clouds from the robot's point of view, after planning.

Fig. 4 (left) shows the relationship between the distance the object moved once placed and the output of the scene discriminator model, after rejecting a small number of outliers (distance $\geq 0.5$ meters). When we compute Pearson's correlation coefficient $r$, we see that for a single random sample $s$, $r_s = -0.147$ with $p_s = 0.161$, and for the final predicted placement $P$, $r_P = -0.278$ with $p_P = 0.006$. Individual initial samples $s$ are shown in blue and optimized placements in red. These results indicate that our scene discriminator learns a metric which correlates with stability, even in challenging, cluttered scenes. Failures of the discriminator are driven by oddly perceived object or environment geometry. When parts of the object geometry are missing, the model is prone to making mistakes, such as placing an object so that it intersects with another object, or placing an object so that it is not properly supported, as in shown in Fig. 4 (right).

### 3.2 Predicate Performance

Table 2 shows the F1 score by predicate on a held-out portion of the dataset containing 3000 examples. Our model correctly predicts every predicate in 69.4% of scenes, vs. 58.5% for the baseline. In addition, we can learn predicates for which we have no baseline, such as *aligned*, on which we have

a true positive rate of 83% ($f1 = 0.43$). Generally, our model classifies different relationships well, matching or exceeding the baseline in every case but one. In particular, our learned model has consistently good *sensitivity*, meaning that we can find valid positions for the object when attempting to plan. Finally, the rule-based approach requires implementing and hand-tuning a range of different predicates; by learning from data, we could in principle scale to a larger number of symbolic relationships. Our results show that even with very unbalanced, "natural" data, we can learn useful classifiers for a range of relationships.

**Baseline implementation**: First, we compute bounding boxes and means from each object point cloud. For the *centered* predicate, we compute $xy$ distance on the table and use a larger threshold that appears in our dataset (2 mm instead of 1 mm). For *touching*, we compute minimum distance between point clouds and threshold it with 2.5mm. For *near*, the threshold is a 5 cm distance. For directional predicates, we apply the exact same rules used in data generation to the computed bounding boxes, as seen in the supplementary materials.

### 3.3 Real World Experiments

Finally, we deployed our system on a Franka Panda robot with an arm-mounted RGB-D camera. We used objects from the YCB object set [35] augmented with toy kitchen objects. We also used various cardboard boxes to force the robot to adapt to changes in scene geometry. We generate grasps using [29] and use RRT-Connect [37] for motion planning. We used unseen object instance segmentation from [12] to determine segmentation masks for different objects, and use a prompt to choose which object to move when performing experiments. We compute standoff poses for each object 10cm above the predicted goal position, and release it 2cm above the predicted pose in order to ensure that we do not press into the table. Figure 1 shows example pairs of before-after images.

To quantify our results in the real world, we conducted a set of experiments with eight different objects. We report results in Table 3. Our placement planner was highly successful at finding manipulation plans with a range of different objects and predicates, including both grasps and placements for the various held-out objects. Crucially, our method is able to find multiple valid solutions for each scene; some examples appear in the supplementary materials. We saw a high rate of grasp execution failures on the real world, which could easily be improved in future work by re-grasping.

| Success Rate | Table Only | | | Table with Large Box | | |
|---|---|---|---|---|---|---|
| | Plan | Grasp | Placement | Plan | Grasp | Placement |
| Sauce Bottle | 3 | 2 | 2 | 2 | 1 | 1 |
| Cookie Box | 3 | 3 | 3 | 3 | 1 | 0 |
| Red Mug | 3 | 3 | 3 | 3 | 2 | 1 |
| Juice Carton | 3 | 0 | - | 1 | 1 | 1 |
| Macaroni Box | 3 | 2 | 2 | 3 | 0 | - |
| Parmesan Can | 3 | 0 | - | 1 | 0 | - |
| Mustard Bottle | 3 | 3 | 3 | 2 | 2 | 2 |
| Large Pitcher | 2 | 2 | 2 | 3 | 2 | 0 |
| Overall (%) | 95% | 75% | 100% | 75% | 56% | 60% |

**Table 3:** Generalization experiments with different objects. All numbers out of three trials. Placement successes conditioned on successful grasps. We see that the largest cause of failures was grasping issues. Planning failed in a few situations with challenging objects that were either very large (YCB pitcher) or very small and hard to grasp (Parmesan can). Placement fails when an object falls off of the box after the planner attempts placement.

## 4 Conclusions and Future Work

We demonstrated the ability for a robot to learn to accurately infer inter-object relations from point clouds of real-world, unstructured environments, which can be used to perform manipulation planning for previously unseen objects. Our results show that a model trained purely in simulation, effectively predicts relations on real-world point clouds of objects not seen during training. Furthermore, our incorporation of a scene-realism discriminator significantly improves performance over the predicate goal predictor alone. In the future, we will add semantic understanding about which objects the robot is interacting with using learned object class and attribute classifiers. We will also expand our method to handle multi-object relations and explore long-horizon manipulation tasks by formally extending our method into a task and motion planner.

## Acknowledgements

Chris Xie was funded by NSF NRI grant IIS-2024057.

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
