# OpenReview forum: "Predicting Stable Configurations for Semantic Placement of Novel Objects"
_robot-learning.org/CoRL/2021/Conference — CoRL2021 Poster_

### Official Review · Reviewer_Tchz · 2021-07-23

**Originality:** Good
**Technical Quality:** Good
**Clarity Of Presentation:** Good
**Impact:** 3

**Recommendation:**

Weak Accept: I recommend accepting the paper, but will not argue for my recommendation if the majority of other reviewers have a different opinion.

**Summary:**

This paper presents a novel integrated approach combining visual perception, logical reasoning, and motion planning of a robot arm for pick-and-place tasks.
Learned through segmented RGB-D data from simulated scenarios, semantic spatial relations between objects are inferred.
A novel discriminator function is also learned to predict the feasibility of placement pose of an object given a semantic goal defined as a relative spatial relation between the query and a secondary anchor object.
The output of this combined visual and semantic module is fed into an of-the-shelf sampling-based motion planner to realize the pick-and-place task by the robot.
This architecture is also shown to be transferable to a physical robot executing such plans in the real-world.

**Issues:**

Some of my suggestions are above (Strengths & Weaknesses). Here, I provide some further improvement points.
Overall, the presentation is good. However, the method section encapsulates both prior work that this paper relies on and the original contribution all together. Either a separation or clear emphasis on what is based on prior work vs new would improve the paper.

The role of the scene encoder is not well motivated/justified and thus might be clarified better.

Supp. material:

Video: Timings need improvement. There is some flickering.

Appendix: There are some typos, unfinished sentences.


**Reviewer Expertise:**

Very good: Comprehensive knowledge of the area

**Strengths And Weaknesses:**

The main strength of this work is the novel combination of the learned (spatial) relational predictor and the physical validity discriminator. The overall system architecture is also well-engineered.

Experimental analysis is detailed and supports the novelty and usefulness claims. The analysis and justification of failure cases are also clear. The real-world transfer of learned components is valuable and encouraging.

In terms of improvement points, I have a few suggestions:
The emphasis on the applicability of the proposed approach for 'unseen' objects is not justified. Because i) it's unclear which type of objects were in the training set vs the leftout 'unseen' ones and how much they resemble/differ from each other w.r.t. their geometries, and ii) the framework relies on segmented point cloud data, 'objectness' seems to become irrelevant or less important.

Decoupling of goal pose prediction/planning and motion control/execution is a critical (design) feature of this work. I can see its benefits, however, many robotic manipulation tasks require sequential decisions that are likely to depend on each other not only in terms of logical validity but also for manipulation planning feasibility. For example, how (with which pose) the robot picks up a tool to use affects the feasible actions for the follow-up steps. Since the authors also suggest investigating long-horizon tasks, the paper can benefit from such a discussion on how the proposed approach can be applied on those scenarios.

It's also unclear how this approach can be extended for multi-object relational predictions and planning.

**Summary Of Recommendation:**

The paper proposes a two-step process for the problem of rearranging objects in an environment by robots.
This strategy comprises learning to predict spatial semantic relations between objects and learning to infer physical validity of placement poses using RGB-D sensory data.
Even though individual components rely on prior works, their combination provides a novel framework for simple pick-and-place tasks given spatial logical goals.
The experimental analysis demonstrates promising initial steps for the applicability of this approach for such problems.
In general, the paper is well-written, and with some revisions and clarifications it can be improved further.

---

> ### Author Response · Authors · 2021-08-24
> **Response to reviewer Tchz**
>
> Thank you for your useful feedback.
>
> > I can see its benefits, however, many robotic manipulation tasks require sequential decisions that are likely to depend on each other not only in terms of logical validity but also for manipulation planning feasibility.
>
> Indeed, we plan on investigating this more in the future as mentioned in the conclusion. We’ve updated the introduction to highlight why we think the current study examines many of the critical points necessary to improve before extending to a full multi-step task and motion planning framework.
>
> > The emphasis on the applicability of the proposed approach for 'unseen' objects is not justified. Because i) it's unclear which type of objects were in the training set vs the leftout 'unseen' ones and how much they resemble/differ from each other w.r.t. their geometries, and ii) the framework relies on segmented point cloud data, 'objectness' seems to become irrelevant or less important.
>
> i) Thanks for this point, we didn’t give significant details about this, instead just referencing the object sets (HOPE and YCB) used. A depiction of all the objects is also included in the supplementary materials. For training, we describe some of the ShapeNet objects in Sec. 2.3; we used bowls, books, bottles, plates, pots, cups, cartons, and several other tools and utensils. These objects were chosen to cover a wide range of small, manipulable objects; none of them appear in our real-world experiments or in our ablation experiments.
>
> ii) While you are right that object semantic identity is not important, as you pointed out in point (i) the difference in geometries is important and object class differences is one proxy for different distributions in object shape that our learned point-cloud feature encoders must generalize across in successfully predicting relations and stable configurations.
>
> > It's also unclear how this approach can be extended for multi-object relational predictions and planning.
>
> While many multi-object relations could be handled by enumerating pairwise-relations (e.g. object x should be left of all objects), you are correct that our method would not directly scale to 3-, 4-, or n-object relations. In cases where the relations always took the same number of objects as input, we could straightforwardly extend our method to learn classifiers with the n relevant objects as input.
>
> However for relations with a variable number of input objects, we would need to use some form of graph or recurrent neural network structure to scale to this problem. This is an active area of ongoing research for us, which we hope to publish in a future work that builds off of the formulation presented here.
>
> > However, the method section encapsulates both prior work that this paper relies on and the original contribution all together. Either a separation or clear emphasis on what is based on prior work vs new would improve the paper.
>
> Yes, unfortunately we didn’t have space for a full separate related work in the paper. We have tried to highlight the contributions of our work and focus on what the new aspects are: the discriminator, planning algorithm, data generation, and to a lesser extent the predicate prediction. We’ve also updated the introduction to more clearly highlight the distinction of our work over previous work in the area.
>
> > The role of the scene encoder is not well motivated/justified and thus might be clarified better.
>
> The point of the object and scene encoders is indeed to allow for generalization over different object geometries, particularly when placing the object at a stable location in the scene. The scene encoder also allows the MDN to capture where the object might be placed in order to avoid large aspects of scene geometry, including clutter that is neither the anchor nor the query object. We added a note to this effect to section 2.1 (line 125).
>
> > Video: Timings need improvement. There is some flickering.
>
> Thanks for pointing this out, we will make sure to clean up the video if accepted to the conference.
>
> > Appendix: There are some typos, unfinished sentences.
>
> We’ve done a thorough edit and revision of the text in the main paper and supplemental materials to fix any issues.

---

> > ### Comment · Reviewer_Tchz · 2021-09-01
> > **Response to rebuttal**
> >
> > Thank you for your clarifications. Due to some of the simplifying assumptions, and concerns over its applicability to multi-step manipulation problems, I cannot upgrade my recommendation but keep it at 'weak accept'.

---

### Official Review · Reviewer_f3r9 · 2021-07-23

**Originality:** Good
**Technical Quality:** Very Good
**Clarity Of Presentation:** Good
**Impact:** 3

**Recommendation:**

Weak Accept: I recommend accepting the paper, but will not argue for my recommendation if the majority of other reviewers have a different opinion.

**Summary:**

The paper proposes a novel approach for predicting stable placements of objects in a rearrangement task. This goal of this task is to rearrange a set of objects given high-level predicates describing their target location such as "above", "near" etc. The paper decomposes this task into multiple subcomponents: (1) Object and scene encoders which project the point clouds of a pair of objects and the scene into a latent space, (2) a relation predictor that predicts a set of fixed spatial binary predicates from the latent space representations the objects and scene, (3) a pose prior used to sample new object locations given a spatial predicate, (4) a scene discriminator used to reject invalid placement proposals, (5) a placement planner to determine a target placement location.

The paper evaluates the approach in a variety of simulation and real-world experiments.

**Issues:**

### Clarity & Presentation

- Introduction: I couldn't really understand from the introduction what the problem the paper wants to solve and how it wants to solve it until reading section 2.
Importantly, the introduction does not explain what the rearrangement task actually *is*, and the argument in L24-29 does not really motivate the task of rearrangement. I would recommend to first explain what the rearrangement task is, why it is challenging, and how this paper decomposes this task. Then it should be explained what alternatives exist (incl prior work) for each subcomponent, and what the choice + contribution of each component of this paper is. I feel that all of this information is somewhere hidden in the introduction, but it needs to be reordered/rewritten in order to guide the reader more gently through the complex matter.

- Figure 2 & 3: I cannot really understand the depictions of the MDN. What do the different colors mean? What do we see here? These figures should be explained in the paper.

- Latent object & scene encoders. First of all, I think it should be stated more clearly what the purpose of these is. If I understand correctly, this component enables the approach to generalize over different object geometries. Second, I'm wondering what role the hard-coded radius delta plays in the generality of the approach.

### Experimental evaluation

Table 1 reports compares the learned and the model-based baseline in terms of specificity and sensitivity, but it is actually hard to tell which one performs better. Eyeballing the numbers, it doesn't seem to me that one can be regarded the clear winner, depending on whether we put more emphasis on specificity or sensitivity. Which one should we give more weight?
(Also, what does it mean for the model-based baseline for "centered" to have 0% sensitivity and 100% specificity?)

I would suggest to use a single metric (e.g. F-Score) to compare the baselines. Furthermore, I would suggest to use bold face for the best performing approach in each row.

### Dataset generation & model-based relationship predictor

It sounds to me that the dataset generation (Section 2.3) and the model-based baseline are large equivalent with the difference that one operates on meshes and the other on partial points clouds. Is this assessment correct? Or does object geometry play a bigger role? I would like to see in the paper a more explicit explanation of the differences.

If that assessment is correct, it seems that the only problem of the model-based baseline is the partial visibility. I would like to see a more elaborate discussion in section 3 if that is the case, or why else the model-based baseline performs (allegedly, see previous point) worse.

Finally, if the partial visibility is a problem, why not add 1-2 additional RGB-D sensors? It should be justified why this is not a solution to the problem, as it would significantly simplify the machinery proposed here to infer spatial relationships.

### Minor issues

- Real world experiments. The paper makes the assumption that object points clouds are fully segmented, but no details apart from a citation of [12] are mentioned in section 3.3. The section mentions that grasping is an issue, but can failures also be attributed to the segmentation?
- L99 "optima" is plura, the singular form is of "optimum"


**Reviewer Expertise:**

Good: General knowledge of the area

**Strengths And Weaknesses:**

## Strengths

The paper addresses a challenging an important problem and proposes a complex and technically sound approach. It provides a large set of both simulated and real-world experiments, and shows the general validity of the approach. The presentation is mostly clear and easy to follow. I like the decomposition of the approach and the inclusion of a scene discriminator as a "critic".

## Weaknesses

1. Presentation and experimental evaluation: The experimental sections contain various studies, and I think it is a good idea to re-order the sections as done in the revision. I now understand what the key result is and why it shows that the proposed method works better. However, the paper would still profit from a clearer guidance for the reader as to what the experimental hypotheses are, how they matter for the final performance, and whether the experiments confirm the hypotheses.

2. Model-free vs. model-based. As discussed in the rebuttal, the dataset generation (Section 2.3) and the model-based baseline are large equivalent with the main difference being that one operates on meshes and one of partial points clouds, and that the model-based baseline requires access to CAD models. The latter difference was not obvious to me, and should be pointed out explicitly in the paper.

3. Justification on decisions for problem formulation. In their rebuttal, the authors argued that multiple cameras are difficult in non-controlled scenarios (I agree) and that collecting multiple viewpoints is too difficult / costly (I don't fully agree). I feel the paper makes some arbitrary choices as to what it expects as given and how it frames the problem - e.g. instance segmentation capabilities are considered as a given, but a dense fusion capability for acquiring and merging multiple viewpoints is not. The paper would profit from a better justification as to why these choices were made.

**Summary Of Recommendation:**

The paper proposes a complex yet mostly sound approach to the complex rearrangement task. The paper has some shortcomings wrt to clarity and the experimental evaluation, in particular in terms of its presentation.

---

> ### Author Response · Authors · 2021-08-24
> **Response to Review f3R9**
>
> Thank you for your detailed feedback, it was very helpful. We believe we can show that our method is substantially better than the baseline in a number of ways, and is both novel and useful; we hope that our revisions make this clear.
>
> > I couldn't really understand from the introduction what the problem the paper wants to solve and how it wants to solve it until reading section 2
>
> We made extensive changes to the introduction to make this clear.
>
> > Figure 2 & 3: I cannot really understand the depictions of the MDN. What do the different colors mean? What do we see here? These figures should be explained in the paper.
>
> Thank you for pointing this out. We expanded our explanation of the MDN figures at line 195 at the end of section 2.2. These show the query object (yellow), anchor (blue), and scene (red).
>
> >  First of all, I think it should be stated more clearly what the purpose of these is. If I understand correctly, this component enables the approach to generalize over different object geometries. Second, I'm wondering what role the hard-coded radius delta plays in the generality of the approach.
>
> Yes, the object and scene encoders allow for generalization over different object geometries, particularly when placing the object at a stable location in the scene. We add this to line 125 in Section 2.1. The hyperparameter radius delta needs to capture the volume around an object; this can be set experimentally, but is not very sensitive.
>
> > Table 1 reports compares the learned and the model-based baseline in terms of specificity and sensitivity, but it is actually hard to tell which one performs better. Eyeballing the numbers, it doesn't seem to me that one can be regarded the clear winner, depending on whether we put more emphasis on specificity or sensitivity. Which one should we give more weight? (Also, what does it mean for the model-based baseline for "centered" to have 0% sensitivity and 100% specificity?)
>
> This is a good point, and we can see why this was confusing. First of all, we would like to clarify that table one does *not* show overall performance, but only shows the quality of the predicate predictor. This is not critically important to the performance of the method as a whole, which is shown in Table 2. We also re-organized the experiments section to emphasize the planner ablation experiments more, which we believe is the more important result to show the importance of our combined approach.
>
> > I would suggest to use a single metric (e.g. F-Score) to compare the baselines. Furthermore, I would suggest to use bold face for the best performing approach in each row.
>
> Done. We’ve used F-score instead in the paper (table 2 now), and moved the previous results to the appendix.
>
> > It sounds to me that the dataset generation (Section 2.3) and the model-based baseline are large equivalent with the difference that one operates on meshes and the other on partial points clouds.
>
> Yes, correct. Our goal was stable placement in clutter even when meshes are not known and with partial occlusions. We assume the specific objects upon which we are testing do not appear in training. We selected objects from different datasets (YCB/HOPE for real-world tests and Table 2, ShapeNet for training).
>
> > Finally, if the partial visibility is a problem, why not add 1-2 additional RGB-D sensors? It should be justified why this is not a solution to the problem, as it would significantly simplify the machinery proposed here to infer spatial relationships.
>
> We discuss this in the text. Multiple RGB-D sensors are not viable in scenarios where you cannot instrument the world, particularly in mobile-manipulation. While it might be possible to build a representation from multiple views, we desire to see how well it performs from only a single view as generating multiple views from a mobile manipulator can be time consuming and is error prone.
>
> > The paper makes the assumption that object points clouds are fully segmented, but no details apart from a citation of [12] are mentioned in section 3.3. The section mentions that grasping is an issue, but can failures also be attributed to the segmentation?
>
> We included an extensive explanation of our unknown-object instance segmentation pipeline in the appendix; there is a large amount of existing work on both grasping unknown objects and unknown object instance segmentation, so we take these capabilities as a given.
> Still, there are many cases where instance segmentation causes issues for grasping and manipulation; we included a discussion of this in section A.5 our appendix, and showed an example where only part of the object is predicted to move. These issues could become more prevalent in multi-step planning, however, replanning and improved state estimation / segmentation over time could improve this. We leave a more thorough investigation to improving segmentation during plan execution as future work.
>
> > L99 "optima" is plural, the singular form is of "optimum"
>
> Thanks, fixed.

---

> > ### Comment · Reviewer_f3r9 · 2021-09-03
> > **Updated review**
> >
> > Thank you for your detailed response and for updating the paper.
> >
> > I think it is a good idea to re-order the sections as done in the revision, I now understand what the key result is and why it shows that the proposed method works better than the baseline. Therefore, I changed my vote to weak accept.
> >
> > However, the paper would still profit from a clearer guidance for the reader as to what the experimental hypotheses are, how they matter for the final performance, and whether the experiments confirm the hypotheses. I also feel that some of the assumptions are a bit arbitrary, e.g. instance segmentation is considered as given, but dense fusion from multiple viewpoints isn't.

---

### Official Review · Reviewer_54xG · 2021-07-24

**Originality:** Good
**Technical Quality:** Good
**Clarity Of Presentation:** Very Good
**Impact:** 3

**Recommendation:**

Weak Accept: I recommend accepting the paper, but will not argue for my recommendation if the majority of other reviewers have a different opinion.

**Summary:**

The paper proposed an approach to manipulate unknown object with vague instructions. The proposed method learns the topology relationship between an anchor object and the unknown object (segmented though).

**Issues:**

The 'unknown objects' in the title could be clarified a bit. e.g. the main focus is to be able to manipulated 'novel' object but still rely on segmentation performance.

**Reviewer Expertise:**

Good: General knowledge of the area

**Strengths And Weaknesses:**

Learning topology and manipulated with vague instructions could improve human robot interactions significantly, and such technique can enable many robotics applications.
However the vague instructions could add more uncertainty and I suspect the affected range surrounding the anchor object might be an sensitive parameters and have strong effects on the performance of the proposed approach.

**Summary Of Recommendation:**

The paper is well organized and easy to follow, and the proposed approach was verified with real world experiments.

---

> ### Author Response · Authors · 2021-08-24
> **Response to reviewer 54xG**
>
> Thank you for your comments.
>
> > However the vague instructions could add more uncertainty and I suspect the affected range surrounding the anchor object might be an sensitive parameters and have strong effects on the performance of the proposed approach.
>
>
> The range around the anchor object is used for predicting object stability (as in the discriminator). We think this is not an extremely sensitive parameter, although it is the “ball query” operation used in PointNet++. This hyperparameter does need to be set properly (in the Pointnet++ backbone, as well as in our code), in that it should be larger than the necessary objects.
>
> If it’s of interest to the reviewers, we could potentially add some experiments looking at this to the camera ready. We didn’t consider this high priority since it’s a fundamental part of pointnet++ models already which have been used successfully in a number of related robot learning tasks such as grasp learning.
>
> > The 'unknown objects' in the title could be clarified a bit. e.g. the main focus is to be able to manipulated 'novel' object but still rely on segmentation performance.
>
> Apologies for this not being clear, you are correct that we do not have a way of recovering from segmentation failures and we make this assumption explicit in the paper. We’ve updated the title to use “novel” instead of “unknown” to alleviate this issue.

---

> > ### Comment · Reviewer_54xG · 2021-09-05
> > **Thanks for the reply**
> >
> > All my questions are answered and I have finished update my review.

---

### Meta-Review · Area_Chair_siRQ · 2021-08-13

**Recommendation:** Accept (Poster)
**Confidence:** 4

**Metareview:**

This paper presents a new approach for predicting stable placements of objects in a rearrangement task, based on ambiguous instructions that specify the spatial relations between the placed objects and the surrounding "anchor" objects. The proposed method decomposes  this task into multiple subcomponents: (1) object and scene encoders, (2) relation predictor, (3) a pose prior used to sample new object locations given a spatial predicate, (4) a scene discriminator used to reject invalid placement proposals, (5) a placement planner to determine a target placement location.

While the paper addresses a challenging problem and proposes a complex and technically sound approach, there were some issues in the paper in its first version, as reported by the reviewers. Mainly, the paper was difficult to understand. The results reported in the experimental evaluation also seemed to be mixed and did not clearly indicate which method works better.

In their rebuttal, the authors updated the paper and resolved most of the mentioned issues, especially in terms of presentation. The authors also added new results and metrics, such as the F1 score, that clearly indicate the advantage of the proposed solution.

---

> ### Author Response · Authors · 2021-08-24
> **Response to meta review**
>
> Thank you for your comments. We have taken the reviewers’ feedback into account and have made some changes to clarify our presentation.
>
> > Mainly, the paper is difficult to understand due to its unclear presentation
>
> Apologies. We have made some changes to improve the quality of the presentation and make our goals clearer. We have highlighted major changes in orange in our updated draft. We wanted to show how we could use learned predicates as a part of a planning and optimization process to find stable placement positions in cluttered scenes, while making as few assumptions as possible about the objects themselves.
>
> > The results reported in the experimental evaluation also seem to be mixed and do not clearly indicate which method works better.
>
> Thank you for the comment. We can understand how the presentation of the results would lead to this conclusion, so we have edited the text to improve this and provide some additional clarity. In fact, we believe the results are unambiguous that our method has strong advantages, when viewed as a whole.
>
> We would like to explain that while, as pointed out by Reviewer f3r9, the learned predicates for some classifiers are slightly worse or comparable to the baselines this is not the main result of the paper. We think instead that the focus would like to draw attention to Table 2 (in the original submission, Table 1 in the updated draft): we showed that with just predicate predictions, it was difficult to find stable positions for objects, and in fact our full algorithm using the scene discriminator substantially outperforms using predicate estimation alone for planning. This is also the purpose of showing a large number of real world experiments where the full algorithm performs well. We’ve reorganized the experiment section to highlight this result first and also improved the abstract and introduction to focus on this result.
>
> As suggested by f3r9, we changed Table 1 (in the original submission, Table 2 in updated draft) and added F1 scores instead of positivity/sensitivity. We also updated the description of these experiments, and have put the sensitivity/specificity from this into the appendix, for the sake of transparency.
>
> We also want to reiterate the “goal” of learning predicates. The main benefit of using learned predicates (Table 1) is that they are better at handling partial occlusions and cluttered scenes, without needing a hand-designed cost function. The logic for the predicates we are capturing is actually fairly complex; we would be satisfied even if we could show our pointnet-based models could capture these relationships on their own.
>
> To conclude, we are happy to see that no concerns were raised about the novelty of our approach or the significance of the planning and real world robot results. We hope this updated clarity of the relational classifier results and improved paper presentations should resolve any concerns of the reviewers.

---

### Decision · Program_Chairs · 2021-09-13

**Decision:**

Accept (Poster)

**Comment:**

This paper presents a new approach for predicting stable placements of objects in a rearrangement task, based on ambiguous instructions that specify the spatial relations between the placed objects and the surrounding "anchor" objects. The proposed method decomposes  this task into multiple subcomponents: (1) object and scene encoders, (2) relation predictor, (3) a pose prior used to sample new object locations given a spatial predicate, (4) a scene discriminator used to reject invalid placement proposals, (5) a placement planner to determine a target placement location.

While the paper addresses a challenging problem and proposes a complex and technically sound approach, there were some issues in the paper in its first version, as reported by the reviewers. Mainly, the paper was difficult to understand. The results reported in the experimental evaluation also seemed to be mixed and did not clearly indicate which method works better.

In their rebuttal, the authors updated the paper and resolved most of the mentioned issues, especially in terms of presentation. The authors also added new results and metrics, such as the F1 score, that clearly indicate the advantage of the proposed solution.